# Information and Parental Consent for French Neonatal Screening: A Qualitative Study on Parental Opinion

**DOI:** 10.3390/ijns9020026

**Published:** 2023-05-03

**Authors:** Julia Pinel, Amandine Bellanger, Carole Jamet, Caroline Moreau

**Affiliations:** 1Departement of Paediatrics, Rennes University Hospital, 35000 Rennes, France; 2Department of Neonatal Special Care, Rennes University Hospital, 35000 Rennes, France; amandine.bellanger@chu-rennes.fr; 3Department of General Practice, Rennes University Hospital, 35000 Rennes, France; carole.jamet@hotmail.fr; 4Biochemistry and Toxicology Laboratory, Rennes University Hospital, 35000 Rennes, France; caroline.moreau@chu-rennes.fr

**Keywords:** neonatal screening, informed consent, genetic testing, parental consent, parental information, health communication, newborn, France

## Abstract

Neonatal screening has excellent coverage in France. Data from the foreign literature raise questions about the informed consent to this screening. The Neonatal Screening and Informed Consent *Dépistage Néonatal Information et Consentement Eclairé* (DENICE) study was designed to assess whether information on neonatal screening provided for families in Brittany allows for informed consent. A qualitative methodology was chosen to collect parents’ opinions on this topic. Twenty semi-structured interviews were conducted with twenty-seven parents whose children had positive neonatal screening for one of six diseases. The five main themes from the qualitative analysis were knowledge of neonatal screening, information received by parents, parental choice, the experience of the screening process, and parents’ perspectives and wishes. Informed consent was weakened by parents’ lack of knowledge regarding choice and the absence of a parent after birth. The study found that more information about screening during pregnancy would be preferable. The information should be repeated and accessible and should make it clear that neonatal screening is not mandatory, but informed consent should be obtained from parents who choose to screen their newborns.

## 1. Introduction

Neonatal bloodspot screening is the most performed screening test in the world. Approximately one in two newborns worldwide benefit from screening for at least one of the screenable conditions [1].

In France, nearly 800,000 children are screened each year, including about 32,500 in Brittany [2]. Neonatal screening is a blood test that takes place between 48 and 72 h of life. There is great variability in the panel of diseases screened between countries [3]. In France, the diseases have been selected according to the criteria of Wilson and Jungner [4] and are fixed by ministerial decree [5,6]. French Neonatal screening detects phenylketonuria, congenital hypothyroidism, congenital adrenal hyperplasia, cystic fibrosis, sickle cell disease, and Medium-Chain-Acyl-CoA-Dehydrogenase deficiency (MCAD).

This screening is a public health issue by improving quality of life and preventing death. Estimates show that since its creation and until 2021 in France, neonatal screening has allowed the detection of approximately 28,000 sick newborns [2].

The mode of parental consent for neonatal screening varies by country. Screening is mandatory in some states of the United States, whereas, in contrast, explicit written consent is required in some European countries.

In France, consent is implied for six diseases screened and must be obtained after mandatory information [7]. The information must be both oral and written, with leaflet delivery before the sample is taken. An information brochure provided by the National Neonatal Screening Center is available to families in all maternity wards [8]. To reduce the false positives rate for cystic fibrosis by increasing the specificity, we associate a genetic analysis of a main mutation of CFTR gene if the immunoreactive trypsin IRT assay is over the threshold. Written consent is only required for this genetic analysis for cystic fibrosis [5,9]. Parents can decline to sign for the genetic analysis, but do not object to the screening. Tacit consent is respected by not objecting. In that case, parents would be contacted if IRT is abnormal.

In 2021, the refusal rate for screening in France was 378 out of approximately 748,000 living children born in France at the time of screening [2]. This low rate of decline can be justified by the public health interest in such screening and its noninvasive nature. However, this raises questions about families’ informed consent for neonatal screening, since no other screening achieves such a result.

Several well-conducted studies on family information, parental wishes, and consent to neonatal screening have led a number of countries to modify their practices [10]. These studies show a general lack of knowledge among families about neonatal screening [11,12]. Thus, 20% to 30% of parents received no information [13,14]. The written information brochure is one of the primary sources of information [13,15] but is only used by parents to a limited extent [11,16]. Two-thirds of parents were unaware of the need for a second test to exclude a false positive [13,16], leading some to mistakenly think their maternity discharge was synonymous with negative screening results [16,17]. The nonmandatory nature of screening was unknown to two-thirds of parents [13]. Some described a situation of fait accompli when screening takes place [18,19,20,21].

Several studies show that not being informed of the possibility of a confirmatory testing and the existence of false positives can amplify emotional reactions and anxiety in parents contacted regarding an abnormal screening result compared with families who were previously informed [22,23,24,25]. This stress experienced can have a long-term impact on the parent–child relationship [23]. With the extension of neonatal screening in France to include several metabolic diseases for which public health interest was demonstrated, the number of abnormal results of the screening will increase [26].

The difficulties detected abroad have also been identified in France by Vailly and Ensellem in their French qualitative sociological study of two Parisian maternity wards [27]. However, the results of studies carried out outside of France cannot be directly extrapolated to the French population. Indeed, France has a different healthcare system organization in terms of neonatal screening, perinatal follow-up, and public health policy. For example, most births take place in the hospital, with a later return home than in other countries [28].

The lack of large-scale French studies involving families and professionals and informed consent for neonatal screening has led to the creation of a mixed methodology study called DENICE. It aimed to simultaneously analyze practices around information and consent for neonatal screening and parents’ experiences, wishes, and expectations. The goal was to obtain the perspectives of both those who explain the relevant information and collect consent and those who consent.

The main objective was to determine if the provided information enables the establishment of an informed consent according to current practices to finally be able to submit improvement proposals for patients and their families.

Our study analyzed the qualitative data obtained through semi-structured interviews with parents who received abnormal screening results for their children. The study focused on the Brittany region in connection with the Regional Neonatal Screening Center of Rennes, which coordinates all the regional screenings.

## 2. Materials and Methods

The DENICE study was a prospective descriptive observational study covering the entire Brittany region. The two parts of this mixed-methods study were conducted independently and in parallel. The first part consisted of a content analysis from qualitative data obtained from semi-structured interviews with parents of newborns screened in Brittany. Only the results of the qualitative analysis are presented below. The other part was an evaluation of professional practices based on our questionnaires using a mixed qualitative and quantitative analysis of the collected data, which is the subject of another publication.

### 2.1. Population and Recruitment

We targeted parents whose children benefited from neonatal screening in Brittany whose one biological result for a disease screened by neonatal testing was abnormal and required the family to be contacted for confirmatory testing and follow-up.

We chose families of newborns with abnormal screening results because the true or false positive tests had an impact on neonatal screening. Indeed, the parent learned his or her newborn diagnostic through newborn screening or had experienced the anxiety of having wrongly suspected a disease in their newborn. Their experiences were similar to those of all families who benefited from the screening, encompassing the entire process concluding in the confirmation or the invalidation of the diagnosis.

We excluded from the study minor or protected adults, non-French speakers because of their inability to carry out the semi-structured interview in French, and the parents who refused to participate in the interview or the recording of the interview.

In the case of an abnormal neonatal screening result, the parents were contacted for a screening confirmatory testing through Brittany’s CRDN caring for their newborn. We prospectively recruited these families thanks to phone numbers provided by the CRDN. In the two months following the abnormal result, parents were contacted by telephone. They had no prior information from their referring physician about our study. During this exchange, participation in the DENICE study was proposed, and the inclusion criteria were checked. Following the telephone contact, an information letter was emailed to those interested in participating in the study before their interview took place.

The participants did not receive any financial incentive or compensation.

### 2.2. Interviews

In order to complete the screening process, the interviews were conducted after the diagnosis had been excluded or confirmed. This was usually less than two months of age. To limit recall bias, the meetings took place within the child’s first five months of age.

Parent interviews focused on the information they received about neonatal screening before and at the time of screening, their consent to the procedure, their experience of being informed of the abnormal screening result, and their expectations regarding information in the field.

Interviews were conducted in person to allow for the analysis of participants’ emotions and to facilitate communication. Both parents could participate in the interview; they were included as two distinct parents in the study but paired in a single family.

The entire interview was recorded to allow for the written transcription of the collected data. Written consent for the audio recording of the interview was obtained on the day of the interview. Oral consent for participation in the study was obtained before the start of the interview after re-explaining the study goals and modalities.

Data collection stopped when the interview data saturation was reached. The average number for studies of this type is 25 interviews.

### 2.3. Data Collection and Analysis

The textual data came from the transcribed semi-structured interviews. The qualitative methodology used was based on the COREQ criteria [29]. The semi-structured interviews were conducted using the interview guide provided in Appendix A.

The age and gender of the participant, their profession, the presence of a child from a previous pregnancy, the type and level of maternity care where the newborn was being taken care of, the age of the newborn at the time of the interview, the presence of neonatal complications, and the child’s health status were collected to allow for subgroup analysis. Parent’s experience of neonatal screening can be influenced by previous births [19]. Thus, from parents, we collected information regarding the presence of other children or their profession if it was related to health.

The analysis of the semi-structured interviews was carried out according to the content analysis method conducted with the assistance of the NVIVO 12 Pro software. An iterative method was used. For this, a thematic grid was pre-established by the deductive method and then completed by the inductive method according to the interview data. We carried out a thematic breakdown and a reformulation of the units of meaning, then distributed and grouped them into subcategories.

We used the double coding technique and performed an encoding comparison to prevent the interpretation of textual data.

### 2.4. Objectives

The main objective of the DENICE study was to assess whether the information given to parents about neonatal screening in Brittany enables informed consent. To this end, we looked for the presence of the following characteristics in the textual data:(1).Quality-delivered information, orally and in writing, by a health professional containing understandable explanations about the purpose of screening, the diseases screened, and the method of communicating results.(2).Free consent, oral and written, for the genetic analysis of cystic fibrosis, after a period of consideration and with the awareness of having a choice.

Depending on several factors, we defined four categories. Therefore, consent can be free and informed, absent, informed but partially free, or free but partially informed.

The secondary objectives of our study were to identify ways to improve the information provided to parents and the practices involved in neonatal screening to meet parents’ wishes and needs for consent. Semi-structured interviews were used verbatim to analyze the wishes and needs expressed by parents concerning the timing of the information, the person providing the information, the timing and modalities of consent, and the content of the oral and written medical information.

The DENICE study was previously approved by the ethics committee of Rennes University Hospital.

## 3. Results

### 3.1. Population

Prospective recruitment took place from March 2021 to January 2022. Interviews were suspended during the health confinements related to COVID-19.

Twenty-seven parents (eight fathers and nineteen mothers) from twenty different families were included in the study and completed the semi-structured interviews. Table 1 presents the characteristics of the parent sample. Five parents had a profession related to health: two care assistants, a nurse, a physiotherapist, and a childcare assistant working with disabled children. One-quarter of newborns were hospitalized in neonatology. The causes of hospitalization were five premature births, one neonatal hypoxic encephalopathy, and two newborn feeding difficulties.

### 3.2. Thematic Analysis

The thematic analysis of the textual data revealed five themes: knowledge about neonatal screening, information received by parents, parents’ choices, the experience of the neonatal screening process, and parents’ perspectives and wishes.

Appendix B is the encoding grid obtained inductively and deductively.

We concluded that the degree of coding agreement between the two researchers, two medical residents, was strong, evaluated by the Kappa coefficient at 0.786 [30].

#### 3.2.1. Knowledge of Neonatal Screening

Parents understood that screening through a blood sample looks for rare diseases in their newborns. Some spontaneously mentioned genetic diseases, and cystic fibrosis was the most frequently cited.

Mother 5:What I understood was that there was a lot of screening grouped together. There was cystic fibrosis, there were other diseases. And that we did this three days after the birth.

Most families understood the method of communicating results. More rarely, parents knew what a false positive was and that additional tests could be necessary in the case of abnormal screening. For instance:

Mother 19:Basically, if it’s negative, you won’t get any news, but if it’s positive, it’s within 6 weeks.

Sometimes, the purpose of the neonatal screening was confused with blood samples taken simultaneously as part of newborn care, which look for neonatal jaundice, an infection, the existence of gastroesophageal reflux, a difficult establishment of breastfeeding, or the antenatal screening of trisomy 21.

Father 2:But no, it was because of the reflux that they did the blood test, I think.

Once the diagnosis was ruled out, some parents mistakenly thought there was still a risk that the newborn had the disease and did not clearly understand the meaning of false positives.

Mother 3:So the pediatrician at the hospital still told us to monitor, especially before these ten years to see if he has any symptoms.

Some families were entirely unaware of neonatal screening. Others were not aware that their child had a follow-up test after an initial abnormal screening.

Mother 1:But he had two screenings in fact? No, but they also did a screening with the platelet while we were there. (addressing the child’s father).

Some parents did not remember the name of the illnesses screened for except when their child had an abnormal screening result. Some mentioned the name of the main organ affected by one of the pathologies.

For some parents, screening was offered only for some newborns. They wrongly believed that the screening test was performed because the care team suspected a disease in their child.

Father 5:I don’t know if it’s specific to babies born with a problem or if it’s systematic.

An explanation provided for the omission of certain information was the passage of time between the interview and screening or the birth context.

#### 3.2.2. Information Received by Parents

A health professional was the primary source of information about screening cited by parents. First, parents mentioned the midwife, either during pregnancy follow-up or at the maternity ward. In the case of hospitalization of the newborn, all parents cited the pediatric nurse. Pediatricians and doctors were mentioned only by two parents.

The pamphlet “3 Days, the Age of Screening” provided parents with relevant information about neonatal screening. The pamphlet allows a parent to access and understand the screening test information when the oral explanation could not take place in their presence. Others have used it as a reminder before their child’s follow-up test. However, eight parents seemed unaware of it, possibly because the pamphlet was lost or not distributed. Finally, some families stated the pamphlet was kept in the Child’s Health Record but was never read.

Mother 12:The midwife gave it to me, and yes it helped me explain to my partner who couldn’t be there. Otherwise I wouldn’t have remembered anything (laughter).

Mother 15:I think what was done well at the clinic was that they gave us a pamphlet. That of course, we don’t read or necessarily read very quickly at the maternity ward, but there was this pamphlet.

Sometimes, the written information provided by this leaflet was overshadowed by internet research results mentioned by many families as a source of information. Their opinions on websites differed. Some used them to become informed in order to deepen their knowledge in consultation with professionals. Others avoided internet research for fear of obtaining biased information.

Mother 13:In any case, during the childbirth preparation course, I was the one who asked the questions because I had already informed myself on the Internet.

For some parents, their knowledge was linked to their own experience. Indeed, some already had the relevant information because of another child who benefited from neonatal screening or because their professional training related to health already addressed neonatal screening. Finally, three parents already knew a person affected by one of the diseases.

Mother 3:Afterwards, I knew a little bit because we already have an older son, so the test had already been done.

Mother 4:In fact, I have a cousin who is very close to me who had a similar problem with her child revealed by the Guthrie test.

To most parents, information about neonatal screening was given after the child’s birth, although four mothers were informed during pregnancy and no fathers were informed. In rarer cases, families were unaware of their child’s screening at the time of sampling or were unaware when they were contacted due to an abnormal result that the first neonatal screening test had even taken place.

Parents judged the content of the information provided as inaccessible because it was too complex. For that reason, several families believed they were insufficiently informed about this test and screening confirmatory testing.

Mother 10:I wish someone had explained it to me a bit more [...], and especially in simpler terms because it can be explained in very technical terms which are difficult to understand.

It was not always easy to inform both parents of the neonatal screening after the child’s birth. For example, a mother could not be transferred to the same hospital as her child, and another had her own health problems after giving birth. Most commonly, fathers could not be present when the information was given in the maternity ward. In all cases mentioned in the interviews, the information session did not take place before birth.

#### 3.2.3. Parental Choice

For parents, signing the form authorizing genetic analysis for cystic fibrosis if the IRT was high was the moment they accepted the neonatal screening after oral consent collection. Only a few parents understood the reasons for written consent. The consent from both parents was taken if both were present; otherwise, only one was collected from the parent who was present.

In parent conversations, we found the principle of autonomy of consent. The parent had the task of choosing in the best interest of the child, and their decision was a result of reflections shared with the other parent, without the influence of professionals. Several families were aware of the possibility of refusing this screening.

Mother 13:It’s not mandatory but it’s strongly recommended so that if there’s a problem, it can be taken care of quickly.

The main motivations that led parents to opt for neonatal screening were the possible benefits to the child in the case of a positive screening result and the reassurance of ruling out the diseases being tested with a minimally invasive examination. For most families who consented, this choice seemed obvious. For others, their trust in healthcare professionals allowed them to choose to screen their child. Parents considered this screening a standard part of newborn care in the maternity ward.

Father 15:We didn’t think much about it in the sense that it seemed like a good thing. And it was pretty obvious to agree to do the research.

Mother 11:We consider it normal. We do it because it has to be done.

However, not all families understood that screening was not mandatory. Therefore, screening was not a matter of choice for those parents. Paradoxically, some did not perceive the possibility of refusal, even though they signed the consent form for molecular biology analysis if the test for cystic fibrosis was abnormal. Some parents accepted neonatal screening without even understanding the meaning of the test. Therefore, their consent was uninformed.

Mother 15:Anyway, Guthrie was mandatory, right? Is that it?

In rare cases and for hospitalized newborns, consent was collected after the sample was taken for screening.

Parental consent took place after the birth of the child. Parents indicated that their ability to receive information and make a choice was impaired during the challenging period of birth and by becoming parents.

The decision-making process can be disrupted. For instance, parents’ mental state could be altered because of fatigue, sleep deprivation, concerns about their child’s health, the stress of becoming a parent, hormone imbalance, painkillers, and the information they must process shortly after birth.

Mother 15:At the maternity ward, everything happens very quickly and then you are tired. It’s true that you don’t necessarily take the time to read or there is information that gets lost.

The period given to families to reflect and make their choices varied from a few hours to a few days. Most families felt this period was sufficient because the choice seemed obvious and easy to make. However, in some circumstances, parents found it too short. They reported being rushed by lack of time, giving consent quickly without much deliberation.

Mother 3:Because it’s true that when you are presented with a done deal. Well, we tell ourselves it’s for our child, we can sign without really realizing what paper we are signing because we are also in the euphoria of the birth.

##### 3.2.4. The Experience of the Screening Process

For most participants, neonatal screening was a routine, mundane act, which made some parents almost forget its importance.

Mother 4:It’s true that I didn’t pay too much attention. I was more in the mindset that he was doing well when he was doing things like heel prick and all that... Since it was a basic test that was done for all babies. I didn’t really ask too many questions.

Parents were surprised by the contact for an abnormal test, but for some, the confusion was compounded as they were unaware of the existence of neonatal screening, the method of announcing results, or the possibility of false positives.

Mother and father 2:The day we got the letter with the child’s name in our mailbox, we didn’t understand what it was.

The surprise gave way to anxiety for many families awaiting confirmation or refutation of the diagnosis. This anxiety was even more pronounced in parents who did not understand the principle of screening or were unaware of it.

Mother 17:Because when you don’t know, when they call you, and tell the mother that her daughter needs to do some tests, you can’t just stay calm.

The information provided by the pediatrician or midwife or, more rarely, even other family members, could provide parents with some reassurance. Despite the negative diagnosis, some parents still had concerns about their child’s health.

Mother 8:Even though she is doing great now. But it’s true that there is always that anxiety behind.

#### 3.2.5. Wishes and Evolutionary Perspectives

Families wanted more accessible content with simple and understandable explanations adapted to the parent. The duration of the information session appeared insufficient to some parents.

Mother 7:Maybe they could take more time to explain things to us. Actually, I find that there are a lot of technical terms used at the maternity ward.

The fact that results were communicated to families only in the case of an abnormal test has had a destabilizing effect. They indicated they would like to have been informed more precisely of the time frame after which the lack of contact would have meant the screening was negative. Parents also wanted to know that an abnormal test requires additional analysis, and that screening can lead to false positives. Finally, when given the relevant information, they would have wanted to know explicitly that these diseases are treatable.

Father 1:After that, we would have liked to know a deadline. How long would it have taken to get results if it was bad. In fact, just to know in general and not to know if there was something. [...] At least, to have a positive or negative response.

Parents wished that neonatal screening was mentioned by healthcare professionals, such as midwives and gynecologists. In the case where the child was hospitalized, the family stated that the pediatric nurse was the ideal person to inform them and address the screening.

Several families suggested written support to supplement the information, especially for those families who do not remember receiving the existing leaflet. This pamphlet should contain more pictures to make it accessible to the majority of people.

Father 11:In writing, I would find it interesting to put this in the childbirth folder of the first days, the classic steps that happen with photos. I even find that for people who do not necessarily have access to reading or something, having something with pictograms can be quite nice.

Several participants wished for the creation of a website to supplement oral information. The website could contain informational videos and parents’ testimonials, which would be helpful initially by providing information on screening, as well as after the announcement of an abnormal test and the diagnostic confirmation.

Mother 13:A website could also be useful. It’s true that nowadays we rely a lot on the internet for information, so it could be helpful.

Families insisted that contact after an abnormal screening should be made by a professional who can answer their questions. This was the most often cited point of dissatisfaction by parents.

Many parents preferred to received information about neonatal screening during pregnancy, before the birth of the child. The information should be given to both parents at different times to allow them a better chance to retain it. However, some parents preferred receiving the information at the maternity ward, as they thought it would worry them during pregnancy.

Mother 1:I think it would be a good idea to have the information in advance because after all it comes all at once and then there can be complications like we had.

Mother 10:And then we say it several times. That the dad is there, it would be really appreciated, I think.

Regarding consent, parents indicated that they wanted an explicit request and needed to know they had a choice. Those who were not cognizant of their right to refuse said they would have accepted to carry out this screening but would prefer to have been aware.

The parental decision must be made after a sufficient period of reflection. The length of this period varies depending on each person. Parents who have given informed consent expressed their satisfaction with how their consent was requested.

Mother 10:Maybe arrive the day before, explain that there will be a blood test and then give time to think about it and arrive the next day with the paper saying here, we talked about it the day before. Now, are you okay with it or not?

Written and verbal consent were preferred. The signing was seen as drawing attention to the fact of making a choice and to the importance of neonatal screening. For only one parent, the decision could be anticipated before birth. The choice should involve both parents.

Mother Family 7:Well maybe by signing, that, people would be interested. Well that’s not what I mean. Maybe they would feel more involved.

### 3.3. The Limits of Informed Consent and Hospitalization of the Newborn

In several situations, giving informed consent was impossible due to the transfer of the newborn to another maternity ward, the parent’s consciousness disorder, or the father’s absence during the stay in the maternity ward. In this case, the other parent consented, which seemed to satisfy the families.

Three parents did not provide informed consent because information and consent collection either did not take place before the screening or never took place. In these cases, the newborns were hospitalized in a neonatal special care unit, and parents were not present at the time of sampling.

Figure 1 presents the number of references to the subtheme of the study depending on whether or not the child of the participating parent was hospitalized in a neonatal unit. The *x*-axis of Figure 1 corresponds to the ratio between the total number of references found in the verbatims of all parents according to their membership in the “neonatal hospitalization for their newborn” category or not, divided by the number of individuals in this category. We divided by the number of parents in each category to make it easier to compare the two different-sized groups. The subcategories concerning the information were negatively impacted in the case of hospitalization. Those related to consent appear similar whether the child went to the neonatal unit or maternity ward.

The other characteristics of the participants did not seem to notably influence the information or consent.

## 4. Discussion

### 4.1. Information

The information provided by professionals enables parents to gain good knowledge about neonatal screening, its benefits, and its implications. Parents’ responses cited the purpose of screening and the method of announcing the results. The results of our study seem slightly better than those obtained by Moody and Choudhry [14] conducted among parents and expectant parents, which could be explained by the fact that families of newborns who tested false or true positive were interviewed.

Many families seemed unaware of having received the brochure “3 Days, the Age of Screening”, may have read only part of it or not read it at all, or have lost it. The same was true for the existing studies [16,27]. This information tool was highly praised by parents, but the challenge was first to make it attractive and accessible and second to distribute it at the right time for parents to benefit from it. The most suitable time seems to be when the parent can read it and look up more information, and not too far from the birth of the future child. To this end, Stewart et al.’s study concluded that the material must not be too long and must contain simple and essential information [31].

### 4.2. Consent

The parent’s decision was partially free or absent for about one-third of the participants. The reasons were a lack of perception of choice due to ignorance that neonatal screening was not mandatory and a lack of information and consent. In the latter case, the newborn care team could not inform parents due to the parents’ circumstances during the maternity stay, or information was not delivered during the newborn’s hospitalization.

Neonatal screening is a test integrated into basic newborn care. Parents favor this screening for the benefit of their child and through their trust in the care teams. Performing this screening test at the same time as a blood test necessary for the care of the newborn can lead to confusion, as shown in the study by Davis et al. [16]. The trivialization of screening may increase the acceptability rate but reduces the importance that parents attach to this test and the explanations given by the professionals on what it involves. Studies have shown that screening automation will influence the poor perception of choice and the belief that it is an insignificant test [18,19,20].

Consent is implied, but written consent is added for genetic analysis in the case of an abnormal screening for cystic fibrosis. We found that many parents associated this signature with the moment of consent. They indicated that the written consent enabled the family to pay more attention to neonatal screening. This is in line with other studies on the subject [16,32]. However, signing did not necessarily imply that the parent understood they were consenting [33].

The optional nature of neonatal screening was not known by all parents. Similarly, Araia et al. and other studies showed that one-third of the parents surveyed did not know they had the option to refuse [13,20,21]. Fagot-Largeault stated that to consent, the subject must express a clear choice, and therefore it is necessary to ask the person [34].

To say screening is optional but recommended would leave parents doubting its necessity. The supposed risk is in decreasing the potential early detection and diagnosis and goes against the interest of newborns. In our study, parents, both aware and unaware of their right to choose, supported a clear announcement that screening is not mandatory. The addition of written consent did not modify the neonatal screening acceptance rate, as demonstrated by Dhondt and Liebl et al. [35,36]. It can be assumed that the participation rate will remain stable even when making parents aware that they have a choice, while emphasizing its benefits for their child. Parents feel more involved when given a choice to make, and this promotes the parent–child–caregiver relationship trust [11,25,37].

The reflection period given to families in order to allow them to choose seemed to suit most parents. Indeed, the decision-making process was based on simple motivations, and the choice appeared obvious and easy for some of them. Other studies offered contrary results, finding that a longer reflection period was desired as a key factor ensuring free decision making without stress and time constraints [18,19,32].

The parent cannot always receive information and consent after the child’s birth. Issues related to the child’s health, basic care, and the mother’s care are the main concerns of parents during the first three days of life. As a result, the parent’s judgment and comprehension abilities may be impaired [38]. Thus, neonatal screening may have seemed insignificant in some cases. The information may be not integrated, and the consent not thought out.

### 4.3. The Experience of Abnormal Screening

Parents were surprised and then worried when contacted about an abnormal neonatal screening test. This revelation was perceived as synonymous with the announcement of bad news regarding their child’s health and seemed unavoidable. However, parents’ lack of understanding and their dissatisfaction were due in part to the lack of knowledge. Other causes mentioned are the possibility of being contacted after leaving the maternity ward and the fact of being contacted by someone who cannot immediately answer their questions. That should be minimized by providing parents with optimal information.

Some parents were still concerned about their child’s health even after the diagnosis was ruled out. Some studies showed that this could lead to parental dysfunctions during the first months of the child’s life [22,23,24]. We did not establish a link between persistent worry, the quality of information, and the consent of the families concerned.

### 4.4. Improvement Perspective

For most families, receiving information and giving consent usually took place after birth. Families expressed a wish for information on neonatal screening to be provided during pregnancy. That would allow families to avoid situations where language barriers, the absence of a parent, or health problems prevent them from obtaining relevant information or giving their consent. Early access to information would allow families time to reflect on their choice, complete their education, inform each other, and read the leaflet. Informing both parents is essential. The French guidelines of good practices require the consent of both parents except when it is impossible [39].

Parents must receive information at different times during pregnancy, after childbirth, and on the day of screening. Repetition allows for better memorization. The International Society for Neonatal Screening (ISNS), as well as several studies, confirm that the most opportune time to inform families is during the antenatal period [11,31,40]. The study by Tluczek et al. [17] on parental options highlighted that the period in which the information is given is more important than how the parents are informed or the person who informs them.

Families identified midwives as the primary resource of information. Preparation for childbirth classes provided group information and thus limited the duration and additional workload caused by antenatal information. In cases where an obstetrician follows the pregnancy, interprofessional collaboration must take place to give them the tools to inform families. This approach has been supported by the American pediatric and gynecology-obstetrics societies [41,42,43].

Pediatric nurses are key players in maternity and neonatal wards. They often perform the neonatal screening blood test in France. However, they also meet families after birth. Thus, their role could be to remind parents of the relevant information and provide any necessary supplementary material, but the initial responsibility for providing parents with requisite information cannot be theirs.

Parents wanted more accessible and straightforward information. The names of the diseases were mostly unknown, apart from cystic fibrosis as the most publicized one. However, parents reported that the organs affected by these diseases seemed easier to remember and were more often mentioned. In addition, they stated that information delivery must specify the maximum delay after which the screening is considered negative. At the same time, it must make clear the possibility of a confirmatory testing in the case of abnormal sampling, the existence of treatment, and that screening is systematic. This aligns with the recommendations by the French professional guidelines for neonatal screening [39].

Parents also indicated that they would appreciate a reference website on neonatal screening. Our study revealed that several parents had already performed online research. Disseminating quality content through a reference website seems necessary and beneficial, as information in a digital format available on the website could be made accessible to the deaf, blind, and illiterate through voice-reading methods. Several countries already have internet resources on neonatal screening [44].

### 4.5. Limitations and Strengths of the DENICE Study

No parent whose child had an abnormal screening for sickle cell anemia or Medium-Chain-Acyl-CoA-Dehydrogenase deficiency agreed to participate in the interview. The parents who were interviewed both experienced the neonatal screening and the entire process leading up to the confirmation or information of the diagnosis of their child. Therefore, there is a memory bias. Thus, the information they retained may differ from that of the entire population of families with screened newborns. However, this bias was deliberately used to ensure that the data obtained would not be too affected by the delay between the interview and the first screening.

We did not include parents to whom information could not be given due to language barriers because this was an exclusion criterion for our study. Nevertheless, informed consent was more delicate for these families, who represent a non-negligible proportion of births in France.

This is the first regional French qualitative study on informed consent for neonatal screening. The results are consistent with other international studies, particularly a British study where consent was also requested from parents. Brittany is a region with an average number of births compared with all regions of France. Collecting qualitative data from parents in person allowed us to analyze their emotional responses. Our study included 30% of fathers, a higher rate than other studies on the subject [24].

According to the data collected on the information content necessary for informed consent and compared with the studies on the subject, Appendix C is a proposed information presentation, which must be adapted to the person receiving the information.

## 5. Conclusions

In most cases, consent to neonatal screening in Brittany is informed. It is necessary to strengthen the perception of choice by families by explaining to them clearly that neonatal screening is not mandatory. Information must be started during pregnancy and should be repeated. Information sources should be multiple and adapted to the greatest number of parents. Linguistic and disability accessibility must be improved, and the content of oral and written information is to be simplified.

Our research highlights situations where informed consent is lacking due to the absence of the parent during the first days of a child’s life. It is essential to change the approach to information delivery in these cases, possibly through antenatal access to information and the identification of a professional responsible for delivering it.

Families had an overall favorable opinion of neonatal screening. Their level of knowledge was good and in line with the guidelines proposed by professionals. Consent was linked to signing the appropriate form for most families. The time for parents to evaluate the information before making their decision was short but seemed sufficient for most families. The practices of health professionals satisfied parents, except for a few cases where the information was not communicated correctly.

The anxiety parents experience when informed of a possible serious illness following an abnormal test can persist beyond the neonatal screening process. The number of false positives is likely to increase with the extension of neonatal screening to new diseases. It would be valuable to instigate further the possible methods of preventing this problem.

More powerful studies in other regions would allow us to deepen the results, especially with new information sources for new diseases that will be screened.

## Figures and Tables

**Figure 1 IJNS-09-00026-f001:**
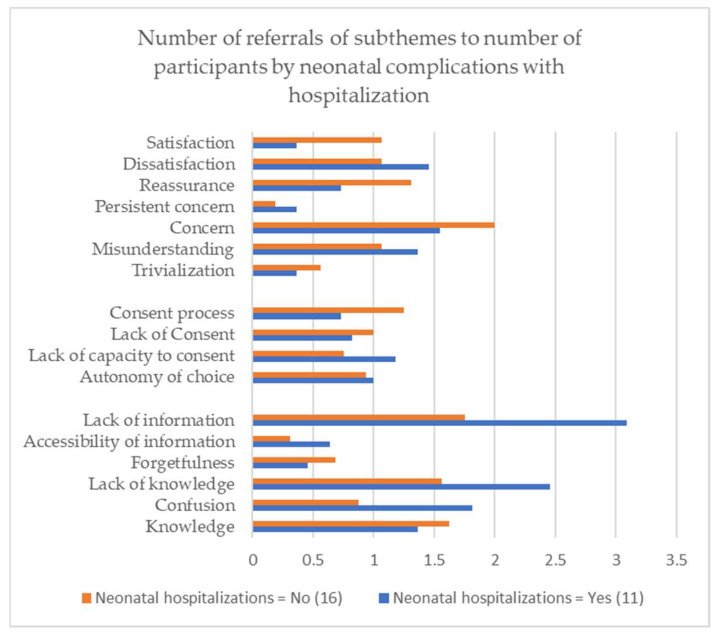
References for subthemes according to the presence of neonatal complications for the child.

**Table 1 IJNS-09-00026-t001:** Characteristics of the population.

Characteristics	Number (%)	Minimum/Maximum	Standard Deviation
Parent	27 (100%)		
Father	8 (29.6%)		
Mother	19 (70.3%)		
First-time parent	16 (59.3%)		
The mean age of parents (years)	33.04	25/45	5.192
Health-related profession	5 (18.5%)		
Maternity			
Public hospital	19 (70.4%)		
Levels of neonatal care *			
1	0 (0%)		
2A	12 (44.4%)		
2B	2 (7.4%)		
3	13 (48%)		
Newborn	20 (100%)		
Newborn hospitalization	8 (29.6%)		
The mean age of children at the interview (months)	3.65	2/5	1.45
True Positive	4 (20%)		
Congenital hypothyroidism confirmed	2		
Cystic fibrosis confirmed	2		
False Positive	16 (80%)		
Congenital adrenal hyperplasia suspected	5		
Congenital hypothyroidism suspected **	6		
Cystic fibrosis suspected **	5		
Phenylketonuria suspected	1		

* Level 1—Special Care Baby Unit for initial and short-term care for newborns. Level 2—Local Neonatal Unit for high dependency 2A without respiratory assistance and 2B with some respiratory assistance. Level 3—Neonatal Intensive Care Unit for complex care. ** A newborn had an abnormal test for both cystic fibrosis and congenital hypothyroidism.

## Data Availability

Not applicable.

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
