# Peer review of "Information and Parental Consent for French Neonatal Screening: A Qualitative Study on Parental Opinion"

_2409-515X, 2023, doi:10.3390/ijns9020026_

Round 1

Reviewer 1 Report

Thank you for submitting this paper which explores the issues around consent and newborn bloodspot screening. Please see detailed comments on the attached.

Suggested changes to the English language style have been included in the attached. Paragraphs should not consist of one sentence. 

Greater clarity is needed regarding existing consent procedure, it is not explicitly clear how this works in terms of it being implied and/or written.

Some points are made in the results without supporting quotes from participants, it would be useful if all points made can be further illustrated with an appropriate verbatim quote. 

Attention is needed to the English language presentation. Many of the comments on the attached contain suggested changes but the whole manuscript will need to be proof read prior to resubmission. 

Author Response

I would like to express my sincere gratitude for taking the time to review my article

Point 1 : Suggested changes to the English language style have been included in the attached. Paragraphs should not consist of one sentence. 

  • We group sentences.

Point 2 : Greater clarity is needed regarding existing consent procedure, it is not explicitly clear how this works in terms of it being implied and/or written.

  • We add a paragraph to explain the French procedure of consent.

Point 3 : Some points are made in the results without supporting quotes from participants, it would be useful if all points made can be further illustrated with an appropriate verbatim quote. 

  • We add some quotes illustrating results. However, we did not put a quotation for each idea in order not to weigh down the text.

Point 4 : Comments on the Quality of English Language

  • We modified everything according to your suggestions.
  • We use the MDPI English editing service for author and they gave us a certificate.

Thank you for your invaluable contribution to this project.

Reviewer 2 Report

The overall manuscript flows smoothly and clearly states its purpose.

How was the X axis for figure 1 developed?  This needs an explanation.  Intuitively, a longer value implies more occurrences, but it is not easily determined if this is a ranked concern, the results of individual Likert scale options or a representation of some fraction of the responses. How did you get values of approximately 0.4 or 1.4? A expansion of the paragraph in the text above the figure may be sufficient.

Section 3.2.2 might be strengthened in the fifth paragraph (Line 256) by the inclusion of some discrete numerical values, such as 4 out of 7 mothers were informed during the pregnancy.

There are minor linguistic issues which resolved from the imperfect 1: 1 correlation of words and common usage between English and French.  The most often repeated one, within this manuscript, is ‘control’, which more typically in English might be ‘confirmatory testing’.  This occurs in multiple places including lines 106 and 501 (as 2 examples).  Similarly on line 188, neonatology is used not as the science of neonatal medicine but implying a neonatal special care or intensive care unit.  Like the prior example, this also occurs in other places such as line 391

In table 1, the accent over the first ‘e’ in phenylketonuria is not needed.

On line 360 consider ‘to be informed’ for ‘inform’. 

Perhaps the sentence involving the lines 461-464 can be rewritten as 2 sentences for ease of understanding since there are too many clauses and ‘and’s.

On lines 489-490, either do not capitalize pediatric, gynecologic-obstetric and societies, or use the names of these organizations such as the American College of Obstetrics and Gynecology (ACOG).

On line 563, ‘the’ is not needed in the form the date is presented.  It could be retained if written as ‘the 23rd of September, 2020’.

In the Appendix A line 572, for better flow of English, consider inserting ‘a’ between am and Pediatrics

In Appendix C, on line 695 change the period to a comma.  This reviewer also suggests, that for ease of understanding, that the order of the components of the lines 696-697 be changed to brain, blood, the lungs, the use of the body's energy resources, or hormones such as those of the thyroid or adrenals.  Then in lines 702-703, a parallel construction should be used so that the disorders are presented in the same order as the involved organ systems which would make this easier for someone to remember.  This reviewer recognizes that the screened disorders may be listed currently as they were historically added to your national screening program.

Reference 27 could be cited in the original form:  Vailly J, Ensellem C. Le consentement au dépistage néonatal ou les sujets ambigus de la génétique. Raison-publique.fr : arts, politique, société, 2012, La bioéthique en débat, http://www.raisonpublique.fr/article528.html with the title translated for the benefit of non-French readers. Finding the reference at the HAL repository site was easy however.

Author Response

I would like to express my sincere gratitude for taking the time to review my article

Point 1 : How was the X axis for figure 1 developed?  This needs an explanation.  Intuitively, a longer value implies more occurrences, but it is not easily determined if this is a ranked concern, the results of individual Likert scale options or a representation of some fraction of the responses. How did you get values of approximately 0.4 or 1.4? A expansion of the paragraph in the text above the figure may be sufficient.

  • We add a paragraph to explain what is the X axis.

Point 2 : Section 3.2.2 might be strengthened in the fifth paragraph (Line 256) by the inclusion of some discrete numerical values, such as 4 out of 7 mothers were informed during the pregnancy.

  • We add more quantitative value to illustrate results of this section

Point 3 : Comments on the Quality of English Language

  • We modified everything according to your suggestions

Thank you for your invaluable contribution to this project.
